# Mathematical Modeling of Remdesivir to Treat COVID-19: Can Dosing Be Optimized?

**DOI:** 10.3390/pharmaceutics13081181

**Published:** 2021-07-31

**Authors:** Jessica M. Conway, Pia Abel zur Wiesch

**Affiliations:** 1Department of Mathematics and Center for Infectious Disease Dynamics, Pennsylvania State University, University Park, PA 16801, USA; 2Department of Biology and Center for Infectious Disease Dynamics, Pennsylvania State University, University Park, PA 16801, USA

**Keywords:** COVID-19, SARS-CoV-2, remdesivir, PK/PD model, antiviral

## Abstract

The antiviral remdesivir has been approved by regulatory bodies such as the European Medicines Agency (EMA) and the US Food and Drug administration (FDA) for the treatment of COVID-19. However, its efficacy is debated and toxicity concerns might limit the therapeutic range of this drug. Computational models that aid in balancing efficacy and toxicity would be of great help. Parametrizing models is difficult because the prodrug remdesivir is metabolized to its active form (RDV-TP) upon cell entry, which complicates dose–activity relationships. Here, we employ a computational model that allows drug efficacy predictions based on the binding affinity of RDV-TP for its target polymerase in SARS-CoV-2. We identify an optimal infusion rate to maximize remdesivir efficacy. We also assess drug efficacy in suppressing both wild-type and resistant strains, and thereby describe a drug regimen that may select for resistance. Our results differ from predictions using prodrug dose–response curves (pseudo-EC${}_{50}$s). We expect that reaching 90% inhibition (EC${}_{90}$) is insufficient to suppress SARS-CoV-2 in the lungs. While standard dosing mildly inhibits viral polymerase and therefore likely reduces morbidity, we also expect selection for resistant mutants for most realistic parameter ranges. To increase efficacy and safeguard against resistance, we recommend more clinical trials with dosing regimens that substantially increase the levels of RDV-TP and/or pair remdesivir with companion antivirals.

## 1. Introduction

In November 2020, the first ACTT-1 study results were released, showing that the antiviral drug remdesivir (commercial name Veklury) had some efficacy in treating COVID-19, specifically in shortening the time to recovery in adults who were hospitalized with COVID-19, and providing evidence of lower respiratory tract infection [1]. At the time, no therapeutic agents had been shown to be efficacious against COVID-19, and these study results were welcome news. Remdesivir had already been granted emergency usage approval by large organizations such as the European Medical Association (EMA) [2] and the US Food and Drug Administration (FDA) [3]. These approvals were granted based largely on evidence that remdesivir successfully prevented disease in rhesus macaques infected with Middle East respiratory syndrome coronavirus (MERS-CoV), which is closely related to SARS-CoV-2, the virus associated with COVID-19 [4], and preliminary ACTT study results showing that remdesivir accelerated recovery from advanced COVID-19 [5].

Despite these positive results, the efficacy of remdesivir remains a topic of debate. A follow-up WHO meta-analysis—for the Solidarity Clinical trial—suggested that remdesivir is not effective [6]. However, questions surrounding study selection and the mathematical approach in this analysis remain, and a more focused meta-analysis of the same data has shown some benefit in patients requiring oxygen [7]. More recent meta-analyses and studies show that remdesivir improves recovery time [7,8] but does not reduce mortality [7]. These are observational studies, but one can also, with some difficulty, investigate the impact of remdesivir on SARS-CoV-2 viral replication in hosts. A modeling study from the French COVID Cohort Investigators and French Cohort Study groups suggested that remdesivir did not improve SARS-CoV-2 viral clearance time [9]. But their sample size was small, and animal [10] and larger human studies [11] do show accelerated viral decay in remdesivir-treated macaques and humans, respectively.

Therefore, questions of whether remdesivir therapy should be abandoned or optimized for use in combination therapy for COVID-19 remain. On the strength of the encouraging, though not definitive, monotherapy studies, multiple studies investigating remdesivir in combination with other therapies, such as dexamethasone [12], are ongoing [13]. Results so far are positive. For example, results from the ACTT-2 study showed that remdesivir in combination with baricitinib, a Janus kinase inhibitor, offered significant improvements in reducing recovery time and accelerating clinical improvement [14].

The major challenge is that remdesivir dosing itself has not been optimized. Study outcomes showing poor efficacy may be a consequence of sub-optimal dosing. At the time of this writing, we are not aware of dose-ranging or dose fractionation studies for more effective RDV regimens. Here, we propose predicting an optimal RDV regimen through pharmacokinetic and pharmacodynamic (PK/PD) modeling. PK/PD modeling is a cornerstone of optimizing drug dosing. We develop a novel PK model to describe the distribution in the body of the remdesivir (RDV) prodrug and its active triphosphate metabolite GS-443902 (RDV-TP) only, since the metabolic pathway leading from extracellular RDV to intracellular RDV-TP is not yet confirmed [15,16,17]. We parametrize the model using data from safety and tolerability investigations reported in Humeniuk et al. (2020) [15].

We further develop a novel PD model that relies on the underlying interactions between RDV-TP and target receptors to predict a drug regimen’s efficacy, as the drug interferes with viral replication. In general, incorporating drug–target binding in models has added to our understanding of drug therapy and was shown to improve quantitative predictions of drug action in antibiotic and HIV therapy [18,19,20,21,22]. Optimizing dosing therefore increasingly relies, not on PK/PD models, but on PK/TE/PD models, where TE stands for target engagement [23].

We use our model to investigate the efficacy of the current drug regimen and to predict a drug regimen with improved efficacy, while being mindful of the possibility of drug toxicity. We then investigate the propensity of the proposed drug regimen to engender and select for drug resistance in SARS-CoV-2. Drug resistance is a major obstacle to delivering effective antiviral or antibiotic therapy for viral or bacterial infections, respectively. In the worst case, we lose effective therapies. For example, amantadanes (amantadine and rimantadine) were very effective in treating influenza A, with efficacy rates of up to 90%, but as of 2020 more than 69% of H1 subtypes are resistant. It is precisely due to this high level of resistance that adamantanes are no longer recommended for the treatment of influenza A [24].

Already, there is documentation of emergent RDV resistant SARS-CoV-2 in an immunocompromised patient being treated with RDV for COVID-19 [25]. This observation matches expectations from in vitro studies [26,27]. Specifically, RDV inhibits the SARS-CoV-2 RNA-dependent RNA polymerase (RdRp, encoded by nsp12-nsp7-nsp8), and mutations in RdRp can decrease sensitivity to RDV. Alarmingly, in vitro, these mutations leave viral fitness largely unaffected [26,27,28]. Thus, as we will show, resistance can easily be selected for by some RDV drug regimens. While at this time there is limited concern that resistant strains will transmit or become dominant, since RDV is administered in hospital only, acquired RDV resistance of SARS-CoV-2 in a host can limit the capacity of RDV to accelerate viral clearance and improve clinical outcomes.

In the following, we describe our model of PK and PD of RDV and its intracellular active metabolite RDV-TP. Referencing estimates for rates of viral replication in the lower respiratory tract (LRT) and upper respiratory tract (URT) [29], we show the efficacy of the current treatment regimen. We aimed to predict an optimal RDV dosing regimen and to discuss the selection of RDV-resistant SARS-CoV-2 variants. We therefore derive from our model an optimal dosing rate and show that, at this rate, one can, at best, achieve RDV-TP concentrations sufficient to fully suppress spread within the LRT, but not the URT. We conclude that RDV will serve best as a companion therapy. Finally, we show the importance of careful dosing by demonstrating our model predictions on the broad regime of a drug regimen that will select for RDV resistant variants.

## 2. Materials and Methods

### 2.1. Pharmacokinetic Model

We develop a pharmacokinetic (PK) model to describe the distribution in the body of the remdesivir (RDV) prodrug and its active triphosphate metabolite GS-443902, (RDV-TP) parametrized using data from Humeniuk et al. (2020) [15]. We focus on these two components because the metabolic pathway leading from extracellular RDV to intracellular RDV-TP is not yet confirmed [15,16,17]. However, there is consensus that intracellular RDV-TP is the active form of the drug [16,17,30]. Examination of the available data suggests nonlinear behavior. Notably, RDV shows biphasic decay, with a rapid drop in concentration following termination of RDV infusion. Further, reportedly the half-life of RDV-TP in peripheral blood mononuclear cells (PBMCs) varies with dose size and duration of infusion, as does the peak concentration (note that, to our knowledge, there are no available longitudinal data on RDV-TP). We are therefore motivated to develop a nonlinear model to describe their dynamics. The model equations are given in Equation (Equation 1), where *R* represents the concentration of RDV in plasma, *P* in the periphery, and *A* represents the intracellular concentration of the active metabolite RDV-TP.
(1)$$\begin{array}{cc} \frac{dR}{dt} & =\frac{q_{0}}{V_{r}}H\left(t-\tau \right)-\left(\delta_{R}+\beta \right)R+kP-d\frac{R^{2}}{D^{2}+R^{2}}\\ \frac{dP}{dt} & =\beta R-(k+\delta_{P})P\\ \frac{dA}{dt} & =\sigma d\frac{R^{2}}{D^{2}+R^{2}}-\mu_{1}A-\mu_{2}\frac{A^{2}}{M^{2}+A^{2}}. \end{array}$$

Remdesivir is infused at a constant rate of $q_{0}=\mathrm{Dose}/\tau$, where τ is the infusion duration. We model the short infusion using a Heaviside function,
$$H\left(x\right)=\left\{\begin{array}{cc} 1, & \mathrm{if}x\geq 0\\ 0, & \mathrm{otherwise} \end{array}\right.$$
RDV in plasma, *R*, diffuses to the periphery at rate β, returns from the periphery at rate *k*, and is eliminated at rate $\delta_{R}$. RDV in the periphery, *P*, is eliminated at rate $\delta_{P}$. RDV in plasma, *R*, can also be absorbed into cells and converted through a series of reactions to the active metabolite *A*. In the absence of definitive reactions and longitudinal data on intermittent steps, we model this process using a Hill function with rate *d* and Hill coefficient 2. *A* is eliminated linearly at rate $\mu_{1}$ and nonlinearly at rate $\mu_{2}$, again modeled with a Hill function with Hill coefficient 2.

We estimate model parameters using the Nelder–Mead algorithm as implemented by the “optim” function in R. Specifically, we use longitudinal RDV concentration measurements in the plasma following single-dose administration [15] in addition to PBMC pharmacokinetic parameters (AUC${}_{\infty }$, C${}_{max}$, C${}_{24}$ and $t_{1/2}$) of RDV-TP in the single-ascending-dose study in Humeniuk et al. [15]. Parameter estimates are provided in Table 1 and details on the fitting are provided in Appendix A. We show how our model fits compare with the data in Figure 1 and Appendix A.

Note that, while we estimate model parameters from single-dose studies, we model a multiple-dose drug regimen. We justify this use by noting that there was no observed accumulation of RDV in the multi-dose study, likely because of its short half-life [15]. There are no data on the metabolite RDV-TP from multi-dose studies, but we note that Humeniuk et al. 2020 [15] reported the accumulation of an intermediate metabolite, and our nonlinear model predictions also show such an accumulation.

### 2.2. Pharmacodynamics Model

To the best of our knowledge, the half maximal inhibitory concentration (IC${}_{50}$) or half maximal effective concentration (EC${}_{50}$) of RDV-TP has never been determined experimentally. We therefore assume that the production of new virions is inversely proportional to the number of occupied binding sites in the viral target, the polymerases producing a nascent RNA chain. The half-life of intracellular RDV-TP is very long, such that the binding should be in equilibrium and the equilibrium binding constant $K_{D}$ is sufficient to calculate target occupancy. In the absence of other data, we use the predicted $K_{D}$ as derived via the binding energy from molecular modeling studies ($K_{D}$ = 3.6 μM, [32]). For the hepatis C virus, the directly measured IC${}_{50}$ of RDV-TP was close to this value (5.6 μM) [33]. Importantly, we here define the target as all potential insertion sites where RDV-TP could integrate into the nascent RNA chain and thereby disrupt the production of a functional viral genome. In reality, this process is very complicated [33]. Here, we use a very simplified approach to be able to work with the available data; for more complicated approaches, we do not have enough reliable parameter measurements yet. Thus, we assume that the antiviral efficacy of RDV-TP is at its EC${}_{50}$ when the intracellular concentration is exactly at the $K_{D}$.

The fraction of occupied target, $f_{occupied}$, is then described by Equation (Equation 2),
(2)$$f_{occupied}=\frac{\left[RDV-TP\right]}{\left[RDV-TP\right]+K_{D}},$$
where [RDV-TP] is the concentration of RDV-TP and $K_{D}$ the affinity as described above. As stated above, we assume here that the production rate of new virions is inversely proportional to $f_{occupied}$. In the context of in-host viral infections, the basic reproduction number, $R_{0}$, is defined as the number of new cell infections induced by a single infected cell, and is proportional to the viral production rate [34]. We therefore assume that the effective viral reproductive number, depending on the drug concentration, $R_{d}$, depends on $R_{0}$ and the amount of free target, as described by Equation (Equation 3):(3)$$R_{d}=R_{0}\left(1-f_{occupied}\right).$$

To suppress viral spread between cells, $R_{0}$ has to be below 1. Thus, the minimal effective fraction of the occupied target is given by Equation (Equation 4),
(4)$$f_{occupied,e}=1-\frac{1}{R_{0}}.$$

### 2.3. Concentrations Selecting for Resistance

Resistant strains have a lower effective affinity, either because mutations lead to reduced binding, or because excision repair removes RDV-TP from the nascent RNA chain. Since the affinity constant $K_{D}$ is inversely correlated to affinity, the effective $K_{D,res}$ of the resistant strain must be larger than the one for the wild-type, $K_{D,wt}$. This results in a lower target occupancy for the resistant than the wild-type (wt) strain at the same concentrations of RDV-TP (Figure 2A).

At the same time, resistant mutants are typically less fit than the wild type and their within-host $R_{0}$ is lowered by $w_{res}$, which describes the relative fitness of the resistant mutant as $R_{0,res}=w_{res}R_{0,wt}$.

Using Equations (Equation 2) and (Equation 3) to obtain the minimal effective concentration for both wild type and resistant strains, we then obtain Equation (Equation 5),
(5)$${\left[D\right]}_{suppression}^{wt}=\left(R_{0}-1\right)K_{D,wt}.$$
This means that less drug is needed when $K_{D}$ becomes small (and therefore drug–target affinity large), and more drug is needed when $R_{0}$ rises (and therefore the virus replicates more). The minimal effective concentration for resistant strains is given by Equation (Equation 6),
(6)$${\left[D\right]}_{suppression}^{res}=\left(R_{0}w_{res}-1\right)K_{D,res}.$$
This latter Equation (Equation 6) also gives the maximal dosing that can select for resistance, that is, the upper limit of the resistance-selection window in Figure 2B.

The lower limit of the resistance selection window is given by the lowest concentration at which the viral production rate and therefore the $R_{D}$ of the resistant strain exceeds that of the wild-type. With $K_{D,res}=K_{D}k_{res}$, where $k_{res}$ describes the decrease in effective drug–target affinity in the resistant strain, this concentration, ${\left[D\right]}_{selection}$, is given by Equation (Equation 7),
(7)$$D_{selection}=\frac{K_{D}k_{res}\left(1-w_{res}\right)}{k_{res}w_{res}-1}.$$
To be selected for at a given drug concentration [D], ${\left[D\right]}_{suppression}^{res}\geq {\left[D\right]}_{selection}$, that is, there must be at least one concentration at which the resistant strain is selected for but not suppressed yet. For a given decrease in effective drug–target affinity, the minimal fitness of a resistant strain that allows resistance to arise can therefore be given by Equation (Equation 8),
(8)$$w_{res}=\frac{k_{res}+R_{0}-1}{k_{res}R_{0}}.$$

### 2.4. Model Validation: Recapitulation of In Vitro Experiments

While the dose–response curve of RDV-TP has never been measured in vitro, there are several publications that investigated the dose–response relationship and EC${}_{50}$ of the parent drug remdesivir in vitro as measured by the inhibition of viral production. This has also been referred to as a “pseudo-EC${}_{50}$” because it does not reflect the molecular mechanism of action. Additionally, remdesivir is thermally unstable and rapidly degraded at 37 ${}^{\circ }$C [35], such that the initial remdesivir concentration is not suitable for obtaining an EC${}_{50}$. As a validation step, we set out to investigate whether our PK model (that describes the metabolization of RDV to RDV-TP and cell entry) together with our PD model can reproduce realistic estimates for the pseudo-EC${}_{50}$ (Figure 2 and Appendix A). It is important to note that our PK model was fitted to RDV-TP levels in PBMCs, which are not primary targets for SARS-CoV-2 but a proxy for intracellular concentrations because they can be easily obtained. EC${}_{50}$s and RDV-TP contents also strongly depend on the cell type [36]. However, our estimate of 0.17 μM is well within the range of reported pseudo-EC${}_{50}$ values of 0.01 μM in isolated human epithelial airway cells, 0.28 μM in Calu cells and 1.65 μM or 0.77 μM in Vero E6 cells [36,37].

## 3. Results

### 3.1. Standard Dosing Does Not Achieve Effective Concentrations

SARS-CoV-2 can replicate in many different body compartments, and its ability to replicate likely differs between those compartments. In non-systemic infections, the replication rate in the upper and lower respiratory tract (URT and LRT, respectively) are most important for understanding disease outcome. $R_{0}$ has been estimated to be 8.5 [4.3–25] for the URT and 27.5 [19.8–42.1] for the LRT [29]. Using Equations (Equation 2) and (Equation 3), we therefore assume that a target occupancy of, on average, 88% [77–96%] in the URT and 96% [95–98%] in the LRT is required to suppress viral replication. Figure 3A illustrates the pharmacokinetics of RDV and RDV-TP under standard RDV dosing (30 min–2 h infusion of a loading dose of 200 mg and 100 mg thereafter [1]). Figure 3B illustrates the resulting target occupancy and indicates the minimal effective target occupancy as described above for both URT and LRT.

Our results thus indicate that the standard dosing regimen of RDV fails to suppress viral replication in either the URT or LRT. In fact, it lowers the viral production rate by an average of roughly 70%, where at least 88% would be necessary to suppress viral replication in the URT in an average patient. However, our results also indicate that the RDV is not entirely ineffective, it is only dosed suboptimally.

### 3.2. Model-Predicted Optimal Dosing Rate

Using our nonlinear model Equation (Equation 1), we can predict an RDV dosing rate that would approximately maximize the intracellular active metabolite RDV-TP, thereby maximizing the efficacy of RDV. RDV data [15] (Figure 1A,B) show that plasma RDV achieves steady-state rapidly after initiating the infusion, and decays very rapidly to low concentrations after the infusion ends. We therefore use the RDV steady-state from Equation (Equation 1), $\;\bar{\;R\;}\;$, to then maximize the amount of RDV ultimately converted to RDV-TP, $d\;\bar{\;R\;}\;/\left(D^{2}+\;\bar{\;R\;}{\;}^{2}\right)$, as a function of the dosing rate $q_{0}=\mathrm{Dose}/\tau$. Details are provided in the Appendix A. Thus, we predict with our model that a dosing rate of approximately 168 mg/h maximizes intracellular RDV-TP, thereby maximizing the efficacy of RDV therapy for COVID-19. Thus, for example, if a 1-h infusion is planned, a total dose of approximately 168 mg, yielding a rate of 168 mg/h, should maximize the drug’s efficacy. For a 2-h infusion or 30-min infusions, total doses of approximately 336 mg or 84 mg, respectively, would again achieve rates of 168 mg/h and thus would maximize RDV drug efficacy. In the following, we use this optimal infusion rate (rounding both duration and dosing) to explore the efficacy of several dosing regimens.

### 3.3. Improved Dosing with RDV Doses That Were Tested in Phase I Trials

Our results indicate that RDV is currently underdosed. We therefore set out to investigate the highest daily dose tested in phase I studies, 225 mg, infused at an optimal rate (80 min). Again, we assessed a five day treatment regimen. We find (Figure 4) that this dosing regimen performs better than than current standard dosing (cf. Figure 3). However, viral replication would also in this case only be suppressed in the URT at peak concentrations. In the LRT, the necessary concentration would never been reached.

### 3.4. Alternative: Longer Infusion

The source of RDV toxicity remains unclear, but side effects that peak during the time of infusion have been described [38]. This might hint that RDV toxicity is driven by peak RDV concentrations in plasma. If this were true, one could keep the peak concentration of RDV in plasma constant but maximize intracellular RDV-TP concentration by extending the infusion time. Here, we investigate which RDV doses infused at an optimal rate would allow viral suppression in the upper and lower respiratory tracts, respectively. We find that 500 mg infused over 3 h would suppress viral replication in the upper respiratory tract (Figure 4C–F). For suppressing viral replication in the lower respiratory tract, 1350 mg RDV would have to be infused overnight (8 h).

At this time point, it is entirely unclear whether RDV toxicity is indeed mediated by peak plasma concentrations and whether such dosing regimens would be safe. However, our results do indicate that RDV therapy could be further optimized if safe and calls for both dose ranging studies and clarification of which metabolites contribute to toxicity.

### 3.5. Avoiding a Potential DR Mutant

Generally, spontaneous resistance mutations can lower the efficacy of antiviral polymerase inhibitors by up to 20× [39]. When resistant mutants carry a fitness cost, their spread can be more easily mitigated. A useful concept for this is avoiding drug concentrations in the resistance selection window (see Methods Section 2.3). The resistance selection window describes antiviral concentrations, which are high enough that the resistant virus spreads better within the host, but low enough that the resistant virus is not eradicated, that is, antiviral concentrations that select for resistance. Figure 5 illustrates this concept with arbitrary resistance mutations that carry a 20% fitness cost and have a 5× or 20× reduced susceptibility to remdesivir, either because of reduced drug–target affinity or because of increased excision repair (the latter would increase the drug–target dissociation rate). We chose a 20% fitness cost for illustrative purposes; the fitness costs of drug resistant mutations depend on the virus and the mode of action of the drug, and can vary significantly [40,41]. However, 20% is within range, observed across multiple viral infections including SARS-CoV-1 [40,41,42] and is consistent with previous theoretical studies seeking to investigate the impact of fitness cost on emergent antiviral resistance [43].

Although there are case reports of resistance, we do not have a good overview of potential RDV-resistant mutations. We therefore assess the minimal fitness that newly emerging resistant mutants can be selected for (Equation (Equation 8)) with a range of reduced drug–target binding. Figure 6A illustrates that, for a within-host $R_{0}$ larger than two, resistance mutations may carry more than 50% fitness costs and yet can be theoretically selected for. This means that there are drug concentrations that are able to select for resistance, even if it is a very narrow concentration range just below the minimal effective concentration for the resistant strain. Importantly, $R_{0}$ is likely much larger than 2; it has been estimated to be 8.5 [4.3–25] for the URT and 27.5 [19.8–42.1] for the LRT [29].

We then go on to assess the maximal fitness of mutants that could be suppressed with the current dosing regimen that results in an intracellular concentration of approximately 10 μM RDV-TP (Figure 6B,C). We find that, for most realistic decreases in binding affinity, the maximal fitness of resistant strains would be below 0.2 in the URT and 0.1 in the LRT. In other words, resistant strains would have to have extremely severe fitness deficiencies for the current dosing to suppress resistance. The highest dose we simulated here—1350 mg infused overnight (8 h)—results in a mean intracellular RDV-TP content of around 250 μM in the second half of the therapy (i.e., between 60–120 h). This would be sufficient to suppress resistance in the URT, but in the LRT, mutants with an intermediate fitness and intermediate decreases in drug–target affinity might arise.

## 4. Discussion

In this study, we developed a PK/PD model that yields treatment efficacy predictions for remdesivir, which is currently the only approved antiviral to treat COVID-19 patients. Our PK model distinguishes itself from the related work by Goyal et al. [10] in its emphasis on the active form RDV-TP, which has a longer half-life than RDV and therefore alters dosing recommendations significantly. Our model predicts that remdesivir treatment efficacy could be substantially improved by (i) optimizing the infusion rate and (ii) increasing the overall dose.

Typical pharmacodynamic models rely on fitting empirical sigmoidal dose–response curves to data obtained in vitro. For remdesivir, this approach is difficult, because remdesivir is a prodrug and has to be converted to its active form, RDV-TP, upon cell entry [33]. Furthermore, remdesivir is very unstable and rapidly decays at temperatures used in cell culture (37 ${}^{\circ }$C), such that, over the time course of an experiment (15–48 h), only a minuscule fraction of the originally used drug concentration is left [35]. For this reason, the measured EC${}_{50}$s from the cell culture have also been termed “pseudo-EC${}_{50}$s” [44]. It is difficult to use these pseudo-EC${}_{50}$s directly in PK-PD models because remdesivir does not follow the same thermal degradation in vivo (rather, remdesivir decay is described by a pharmacokinetic model) and because diffusion barriers and the necessary metabolic steps of drug activation complicate the relationship between extracellular remdesivir and intracellular RDV-TP. Thus, using only plasma remdesivir levels coupled to this pseudo-EC${}_{50}$ will likely give inaccurate results. Therefore, we employ a pharmacokinetic model to predict the intracellular concentration of RDV-TP and then use RDV-TP target engagement calculated based on a predicted affinity constant from structural studies [32]. Although this approach makes multiple assumptions, we find that our model accurately predicts EC${}_{50}$s measured in cell culture using extracellularly supplied initial remdesivir concentrations.

The pharmacokinetic/target engagement/pharmacodynamic (PK/TE/PD) models that we employ offer a complementary approach to models employing allometry and mouse models. It is common practice to extrapolate from mouse models to humans, and this approach is very helpful if the target tissue is not easily accessible in human patients (e.g., the lungs) but can be obtained from animal models. Indeed, this approach was taken in Hanafin et al. [45], aiming to identify improved RDV dosing strategies. Their allometry-based modeling predicted that intravenously-delivered RDV could not achieve 90% maximal inhibitory concentration, as measured in vitro, of unbound remdesivir in human plasma. While the strategy is common and sound, drugs that undergo extensive metabolism, which is the case for RDV [15,16], can override the effect of size in the simple scaling of drug doses [46]. In addition, reliable allometric scaling to predict human doses typically involves more than one model species [47]. We conclude that our PK/PD model, relying on human pharmacokinetic measurements of RDV and RDV-TP, should deliver more accurate predictions.

However, the EC${}_{50}$ alone is not enough to predict optimal therapy, since it might be necessary to achieve very high levels (not only half-maximal) to suppress viral spread within a host. A typical target would be achieving 90% viral suppression (EC${}_{90}$). Here, we use the efficiency with which SARS-CoV-2 spreads through the body (within-host $R_{0}$) to determine how much viral replication has to be suppressed to clear the virus. While we expect that the EC${}_{90}$ is sufficient in the upper respiratory tract where the virus spreads less quickly, we expect that 96.5% target binding and therefore viral suppression are needed to inhibit viral replication in the lower respiratory tract.

Based on our pharmacokinetic model, which is calibrated to RDV-TP concentrations in blood (PBMC) and not lung tissue, we expect that the current standard dosing regimen fails to suppress viral replication in both the upper and the lower respiratory tracts. Lung tissue concentrations are likely even lower, such that our estimates are conservative [45].

Nevertheless, our results do not indicate that remdesivir is ineffective; they indicate that the current dosing has room for improvement if toxicity allows. To the best of our knowledge, it is currently unclear whether RDV toxicity is mediated by RDV itself or any of its metabolites, and whether this toxicity is correlated to peak concentrations, average exposure or time by which the concentration exceeds a certain threshold. In addition to hepatotoxicity, side effects that peak during the time of infusion have been described [38]. Further studies are required to establish every possible correlation between dose, pharmacokinetics and tolerability.

The observation that there is room for improvement in the dosing regimen is in line with clinical trials that report a shorter time to recovery [1,48], newer meta-analyses that support a slightly reduced mortality [7] and recent findings that the viral load declines more quickly in patients receiving remdesivir [11]. We therefore would like to argue specifically for more dose-ranging and dose-fractionation trials. To minimize toxicity, it would also be important to understand whether remdesvir itself or any of its metabolites drive toxicity, and whether this toxicity correlates best with peak concentrations (C${}_{max}$) or average exposure (AUC).

Remdesivir resistance mutations have been characterized both clinically and experimentally. Unfortunately, one mutation conferring mild resistance does not carry any fitness cost in vitro [26,27,28]. If this mutant has no fitness costs in all relevant environments in the host and during transmission, we would expect it to arise quite frequently in remdesivir treated patients and ultimately to spread between patients, irrespective of mitigation strategies. Resistance mutations without fitness costs in vitro have been described in other anti-infectives [49], but frequently they do not spread epidemiologically, hinting at fitness costs in vivo and/or during transmission. Mutations conferring higher levels of resistance to remdesivir but a significant (but unspecified) fitness cost in vitro have been described already before the pandemic in the very closely related MERS-CoV virus [50].

Our model predicts that the current standard regimen strongly selects for resistance to remdesivir for a vast majority of the credible parameter spaces. We would therefore argue for increasing efforts to find potential resistance mutations in patients treated with remdesivir. Combining remdesivir with one or more other antivirals would not only improve outcome, it likely would also reduce resistance evolution [18,19,20,21,22]. This might be especially important if remdesivir is used as a template for designing new compounds with increased efficacy [51]. Resistance against older, less effective anti-infectives often provides a stepping stone to resistance against improved but similar compounds [52].

Taken together, our work indicates that remdesivir is expected to be mildly effective and therefore a useful companion drug in COVID-19 treatment. However, dosing should be improved and/or antiviral companion drugs should be employed to safeguard against resistance.

## Figures and Tables

**Figure 1 pharmaceutics-13-01181-f001:**
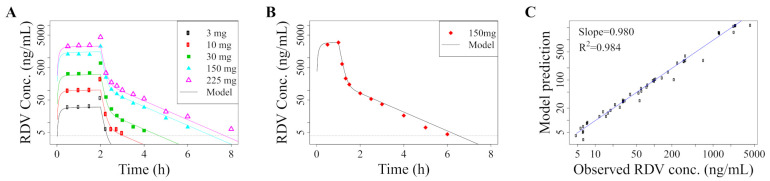
Median RDV plasma concentration from single-dose experiments [15], with different RDV dose sizes and infusion durations, compared to our pharmacokinetic model prediction (Equation (Equation 1)) simulated in R [31] given (**A**) 2-h infusions of 3 mg, 10 mg, 30 mg, 75 mg, 150 mg, 225 mg and (**B**) 1-h infusion of 150 mg. (**C**) Linear regression comparing the model predictions with the data to demonstrate how well the model explains the data. Data from 75 mg dosing excluded in the fitting, see Appendix A for details and Appendix A for comparison. For cases where Humeniuk et al. (2020) [15] give plasma concentrations following multi-dose administration, we use the first dose only.

**Figure 2 pharmaceutics-13-01181-f002:**
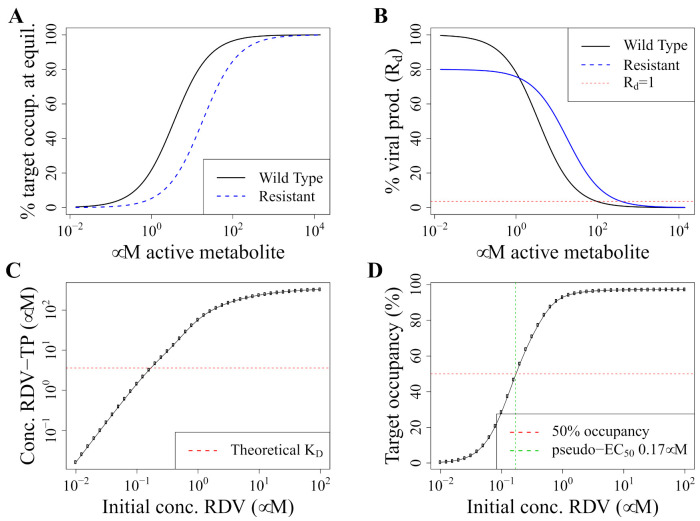
Pharmacodynamic model. (**A**) Target occupancy at equilibrium depending on RDV-TP concentration as calculated with Equation (Equation 2). We assume that a resistant strain has a five-fold reduced $K_{D}$. (**B**) Resulting viral production rate (and therefore $R_{d}$, see Equation (Equation 3)) relative to the rate in the absence of drugs. Here, we assume the same parameters as in (**A**), and additionally set a fitness cost for the resistant strain of 20%. (**C**,**D**): Model validation by simulating “pseudo-EC${}_{50}$” for remdesivir in vitro. This figure shows simulations of in vitro experiments to determine the half-maximal suppression of viral production (EC${}_{50}$) in cell culture when supplying RDV to the medium of cells grown in culture (i.e. extracellular RDV). (**A**) Intracellular RDV-TP concentrations depending on the extracellular RDV concentration (black line) obtained with our pharmacokinetic model described in Section 2.1. The dashed red line indicates the $K_{D}$ of RDV-TP to polymerase binding. (**B**) Target occupancy depending on extracellular RDV concentrations obtained with Equation (Equation 2) (black line). The dashed red line indicates 50% target occupancy. The dashed green line indicates the EC${}_{50}$ as obtained by fitting a log-logistic function to the calculated data shown in this figure with the R package “drc”.

**Figure 3 pharmaceutics-13-01181-f003:**
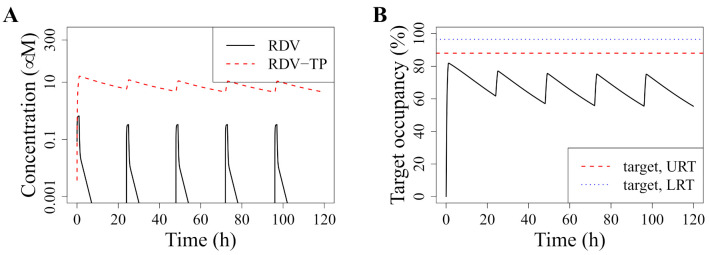
Pharmacokinetics and target occupancy with standard RDV dosing. (**A**) Shows the simulated pharmacokinetics for standard dosing of RDV, i.e., once daily 200 mg loading dose, 100 mg thereafter, infused over 1 h (recommended range 30 min to 2 h). We simulate five days of treatment (time given in hours). The concentration of RDV in plasma (black line) and RDV-TP in blood cells (PBMCs, red line), is given in μM. (**B**) Shows the resulting target occupancy as obtained from the intracellular RDV-TP concentration in (**A**) and Equation (Equation 2). The red dotted line indicates the minimal effective target occupancy for the LRT, and the green dotted line the same for the URT. Both were calculated with Equation (Equation 3).

**Figure 4 pharmaceutics-13-01181-f004:**
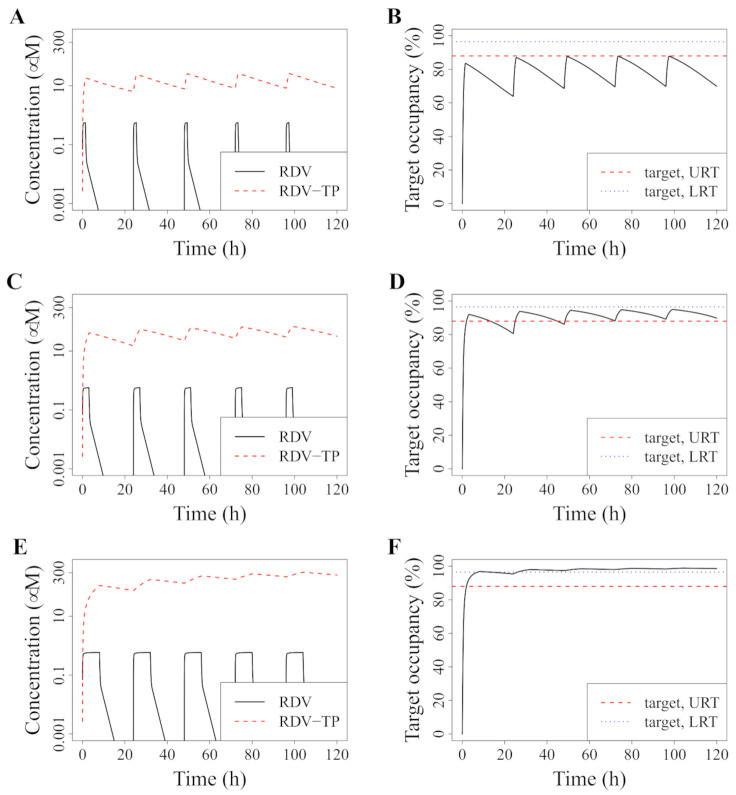
Pharmacokinetics and target occupancy with alternative dosing regimens. (**A**,**B**) 225 mg infused over 80 min. (**C**,**D**) 500 mg infused over 3 h. (**E**,**F**) 1350 mg infused over 8 h. (**A**,**C**,**E**) show the simulated pharmacokinetics. We simulate five days of treatment (time given in hours). The concentration of RDV in plasma (black line) and RDV-TP in blood cells (PBMCs, red line), is given in μM. (**B**,**D**,**F**) show the resulting target occupancy as obtained from the intracellular RDV-TP concentration on the left hand panel and Equation (Equation 2). The red dotted line indicates the minimal effective target occupancy for the LRT and the green dotted line the same for the URT. Both were calculated with Equation (Equation 3).

**Figure 5 pharmaceutics-13-01181-f005:**
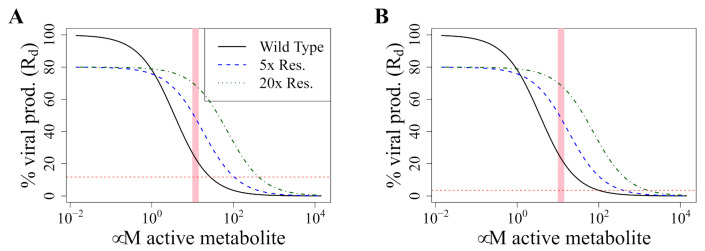
Resistance selection window for upper and lower respiratory tracts. (**A**) Resistance selection window for the upper respiratory tract ($R_{0}=8.5$); (**B**) resistance selection window for the lower respiratory tract ($R_{0}=27.5$). In both figures, the *x*-axis shows the intracellular RDV-TP concentration on the *x*-axis and the relative viral production rate (and therefore $R_{d}$) on the *y*-axis (as obtained with Equation (Equation 3)). The solid black line shows the wild-type, the dotted lines the arbitrary resistance mutations with a 20% fitness cost and a 5× (blue) or 20× (green) increased $K_{D}$. The pink shaded area shows the intracellular RDV-TP ranger measured in phase I trials [15]. The red dotted line marks when $R_{d}$ would become 1 for the upper (**A**) and lower (**B**) respiratory tracts.

**Figure 6 pharmaceutics-13-01181-f006:**
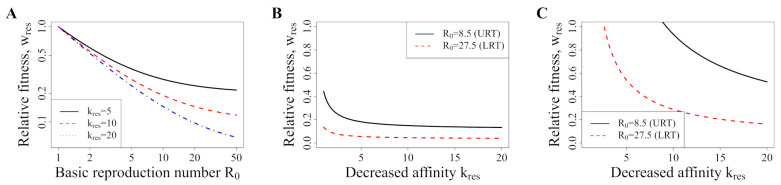
Fitness requirements for selection of resistant mutants. (**A**) Shows the minimal resistance with which newly emerging resistant mutants can be selected for, dependent on the baseline $R_{0}$ in a patient, as obtained with Equation (Equation 8). The *x*-axis shows $R_{0}$ (estimated between 4.3 and 42.1 depending on body compartment and patient). The *y*-axis shows the relative fitness costs. (**B**,**C**) show the minimal fitness with which newly emerging resistant mutants can be selected for, depending on the decrease in drug–target affinity, as obtained with Equation (Equation 7). The *x*-axis shows the parameter $k_{res}$, which describes the factor by which drug–target binding is decreased by a resistance mutation, the *y*-axis shows the relative fitness. The black line shows results for the upper respiratory tract and the red line for the lower respiratory tract; (**B**) shows the results for the standard treatment; (**C**) Shows the results for a 1350 mg/8 h infusion.

**Table 1 pharmaceutics-13-01181-t001:** Pharmacokinetic model parameter estimates.

Parameter	Description	Units	Value
Remdesivir			
Dose	Total mass of RDV infused	mg	varies
τ	Duration of infusion	h	varies
$V_{r}$	Apparent volume	L	6.09
$\delta_{r}$	Elimination rate	per hour	4.16
*k*	Periphery-to-plasma RDV concentration	per hour	0.13
	transition rate		
β	Plasma-to-periphery RDV concentration	per hour	4.42
	transition rate		
*d*	Nonlinear plasma RDV to RDV-TP	mg/L/h	31.57
	(intracellular GS-443902) transport and		
	activation rate		
σ	Conversion of concentrations from mg/L	mol/mg	1
	to μM		
*D*	RDV concentration yielding 50% of the	mg/L	3516.27
	maximal transport/activation rate		
Remdesivir concentration in the periphery			
$\delta_{P}$	Elimination rate	per hour	1.55
RDV-TP (Intracellular GS-433902)			
$\mu_{1}$	Linear elimination rate of RDV-TP	per hour	1.55
$\mu_{2}$	Nonlinear elimination rate of RDV-TP	μM/h	$8.38\times 10^{-3}$
*M*	RDV-TP concentration at which the	μM	17.19
	nonlinear elimination rate is 50% its		
	maximum		

Note: The parameters β and *k* also implicitly include the transition with apparent volumes since the ODEs represent dynamics of drug concentrations. That is, $k=\left(\mathrm{drug}\mathrm{transport}\mathrm{rate}\right)\times V_{p}/V_{r}$.

## Data Availability

Not applicable.

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
