# Peer review of "Mathematical Modeling of Remdesivir to Treat COVID-19: Can Dosing Be Optimized?"

_pharmaceutics, 2021, doi:10.3390/pharmaceutics13081181_

Round 1

Reviewer 1 Report

The manuscript of Conway and our Wiesch dealt with the dose optimization of remdesivir by adopting a computational model approach. The latter identified an optimal infusion rate of the drug that could enable the maximal efficacy of the drug. Interestingly, the model may manage the intracellular active metabolite of remdesivir. This fact significantly increase the interest for the model, because insufficient concentrations of the triphosphate metabolite may result in the selection of resistant clones. I found the reading of the manuscript interesting, and I have listed my comments below.

Lines 226-7. The authors stated that “we assume the same parameters as in (A), and additionally set a fitness cost for the resistant strain of 20%”. Is there a reference to support this choice?

Lines 314-317. The identified infusion rate can maximize the conversion of RDV into its active metabolite. Is this completely true when comparing three different dosages (i.e., 84, 168 and 334 mg/day)? An explanation is due.

Although a simulated dose of 1350 mg/day infused in 8 h seems more effective in reducing the risk of mutant selection, further studies are required to establish every possible correlation between dose, PK and tolerability. This fact  could be stressed.

MINOR

The introduction is pertinent and effective in giving a general overview of the study to the reader, but I suggest to reduce the length of this section for example moving the comparison with other PK models into the discussion. Moreover, some sentences appears in both introduction and methods sections (lines 68-69 and 136-137).

Page 4: it is likely that some symbols are lacking (i.e., lines 154-5, 158).

Lines 207-208. Which is the meaning of the sentence?

Author Response

Dear Editors,

Please find below our point-by-point responses to reviewer comments. Responses are italicized.

Sincerely,

Jessica M. Conway and Pia Abel zur Wiesch

Reviewer 2 Report

Review comments

In this study, the authors developed a PK/PD model that allows predicting the treatment efficacy of remdesivir. And the model predicts that remdesivir treatment efficacy could be substantially improved by i) optimizing the infusion rate and ii) increasing the overall dose. The research is innovative and practical,

Major comments and revision suggestions as follows,

  1. The authors should be provided a list of abbreviations used in the manuscript, and use the full name when it first appears.
  2. The logic of the introduction is poor and not concise enough, so it is recommended to be simplified.
  3. In Figure 3, after each injection of remdesivir, the concentration of remdesivir first increases and then decreases. During the decline, the decline rate of remdesivir is first fast and then slow. What is the cause?
  4. Many details need to be noted, for example, the order of the Figure is all uppercase (Figure 3A, Figure 4A...), but lowercase (Figure 3a, Figure 4a...) is used in both the Figure legends and the manuscript.
  5. The font and size of the formula should be consistent, and each formula should be numbered.

Author Response

(The authors gave the same response as above.)

Reviewer 3 Report

This is a well-prepared manuscript that deals with an important topic. But, some concerns need to be addressed to fit for publication:
1.    The title needs to be representative of the study as the authors should clarify the type of the study as a predictive study using a computational model and clarify the type of program used.
2.    Abstract: add the full term of EMA and FDA.
3.    Introduction should end with the aim of the study. Thus, transfer the last paragraph (lines 122-129) to the discussion or the conclusion sections.
4.    In the results section: the results should be shown only without discussing it. Several paragraphs should be transferred to the discussion section e.g. lines 336-344.

Author Response

(The authors gave the same response as above.)
